# The Sacrificial Ritual and Commissioners to the South Sea God in Tang China

**Yuanlin Wang**

College of Humanities, Guangzhou University, Guangzhou 510006, China; twyl@gzhu.edu.cn

**Abstract:** Previous studies on the Nanhaishen Temple 南海神廟 (Temple of the South Sea God) in Guangzhou in the Tang dynasty focus mainly on the South Sea God as the patron of the Maritime Silk Road, without thoroughly discussing the state ritual and the sacrificial right of the Tang government. This paper illuminates five new points concerning the ritual. First, the sacrificial ritual to the South Sea God developed from the suburban rituals in previous dynasties into both forms of suburban and local rituals, which was also categorized as the medium sacrifice among the three major sacrifices in the state ritual system of the Tang dynasty. Second, the first commissioner who was sent by the central government to perform the sacrificial ritual to the South Sea God was Zhang Jiuling, and henceforth the temporary assignment of court officials to the ceremonies became institutionalized. In the tenth year of Tianbao (751), the South Sea God was entitled Guangliwang 廣利王 (King Guangli), and the commissioner sent on this mission was Zhang Jiuzhang, Zhang Jiuling's third younger brother, rather than his second younger brother Zhang Jiugao as seen in some records. Third, most of the commissioners were dispatched by the central government in the early Tang, and therefore the sacrifice to the South Sea God was related to the state ritual system; but in the late Tang local officials became dominant in the ritual ceremonies, and thus good harvests and social stability in the Lingnan region became the major concern of the sacrifice. Fourth, the legend that the Buddhist Master Xiujiu 休咎禪師 took over the temple and accepted the South Sea God as his disciple reflected the reciprocity between Buddhism and the South Sea God belief. Last but not the least, the sacrificial ceremonies to the South Sea God established in the Tang dynasty and performed by the officials of both the central and local governments had a significant influence on the ritual in the following dynasties.

**Keywords:** South Sea God; state sacrificial ritual; Zhang Jiuling; Zhang Jiuzhang; Zhang Jiugao; Tang dynasty; Buddhism

## 1. Introduction

The Nanhaishen Temple 南海神廟 (Temple of the South Sea God) is one of the best-preserved temples that enshrine the spirits of the four seas in China as its location has not changed throughout the various dynasties since the 14th year of Kaihuang in the Sui dynasty (594). It is, thus, listed as a national cultural relic for further preservation. Since the South Sea God blesses people with safe voyages, a lot of scholars have studied the temple from the perspective of its status as a significant historical relic along the ancient Maritime Silk Road (Huang 2005; Huang and Yan 2011; Qiao 2015). However, similar to the designation of *yue-zhen-hai-du* 嶽鎮海瀆 (sacred peaks, strongholds, seas, and waterways),[1] the Nanhaishen Temple mainly served as a display of the sacrificial right and jurisdictional right of the dynasty from the perspective of state ritual. As a matter of fact, the local and central governments of all dynasties dispatched officials to perform the ritual to the South Sea God. Scholars have studied this topic (Wang 2006), but there are still many questions open for discussion: How were the suburban sacrifice and the local sacrifice to the South Sea God performed in the Tang Dynasty? Were there any differences in sacrificing to the deity in Guangzhou between the early and the late Tang? How did the ritual commissioners of the Zhang brothers play their role in this regard? How was the South Sea God related

to Buddhism? What were the influences exerted by the sacrificial ritual to the South Sea God in the Tang dynasty upon the following dynasties? Taking these questions as points of departure, I aim to figure out what roles the sacrificial ritual of the South Sea God played at the national and local levels in the Tang dynasty, how the central and local officials officiated the ceremonies, and what legacies such a sacrificial ritual in the Tang dynasty left behind for the future generations.

## 2. Suburban Sacrifice and Local Sacrifice to the South Sea God in the Sui Dynasty

The Chinese sacrificial rituals to the renowned mountains and waters correlated with the development of the dynasties. The state sacrifice to rivers and seas, as well as to mountains and hills, begins with religious belief, geographical knowledge, and jurisdictional legitimacy. Some of the mountains and waters were not necessarily in the territory of the state, thus, the rulers offered sacrifices at the suburbs of the capital to worship all the gods and spirits. In this sense, suburban sacrifice was only a symbolic means in the state ritual culture, and what really mattered was the designation of the renowned mountains and waters that could demonstrate the power and territory of the state. In the *Shangshu* 尚書 (Book of Documents), we can find the terms *sihai* 四海 (the four seas) (Kong and Kong 2000, 6.197, 204), *nanhai* 南海 (the South Sea) (ibid., p. 191) and others, and the territory then stretched into infinity. These seas were often used by the rulers of the Warring States period (770 BEC–221 BEC) to demonstrate their sovereignty, so a Sihaici 四海祠 (Shrine of the Four Seas) in Yongzhi 雍畤, which was at the suburbs of the capital of the Qin state (present-day Fengxiang, Shaanxi), was simply a nominal venue for sacrificial ritual rather than a display of jurisdictional and sacrificial rights claimed by the forthcoming unified regimes. It was not until 61 BCE that Emperor Xuan of the Han dynasty established the state ritual system of sacrificing to the five sacred peaks (*wuyue* 五嶽) and four waterways (*sidu* 四瀆), and then the religious and judicial authorities came into being (Jia 2021, p. 319). During the late Western Han dynasty (206 B.C.–A.D.24), suburban sacrifice was the main ritual, though a Haishuici 海水祠 (Shrine of Seawater) was built by the local government in Linqu 臨 (present-day Linqu, Shangdong) (Ban 1962, 25.1243–47; 28.1585). Wang Mang 王莽 reinvented the sacrificial scheme by associating the heaven (*tian* 天) with cosmos, and the earth (*di* 地) with geography according to the belief that "the heaven is like the round mound while the earth is like a square" 圜丘象天, 方澤則地 (ibid., 25.1266), in which "the earth" refers to Tiantan 天壇 (Heaven Altar) at the southern suburbs of the capital while "square" refers to Fangzetan 方澤壇 (Square Altar) at the northern suburbs. It then became the standard ritual of sacrificing to heaven and earth at the suburbs of the capital Chang'an 長安, where the sea gods were sacrificed to at the second grade. The emplacement of worshiping heaven and earth was relocated to the suburbs of the capital Luoyang 洛陽 in the early Eastern Han when the gods of the four seas were also sacrificed to at the second grade (Fan 2000, 97.3160). All in all, sea gods were sacrificed to at the second grade as the main ceremony was offered to the heaven at the Circular Mound Altar at the Southern Suburbs (*nanjiao yuanqiu* 南郊圜丘) and to the earth at the Square Altar at the Northern Suburbs (*beijiao fangqiu* 北郊方丘) during the Han dynasty.

During the Eastern Jin dynasty (317–420), the spirits of the four seas were only sacrificed to at the second grade with a monumental statue at the altar of sacrificing to the earth at the northern suburbs of the capital city Jiankang 建康 (present-day Nanjing). From the 11th year of Tianjian (512) of the Southern Liang dynasty onward, the number of monumental statues was increased to four to sacrifice to the East Sea, the South Sea, the West Sea, and the North Sea. These spirits of the four seas were also named and worshiped in the rest of the dynasty and throughout the Northern dynasty (386–581) (Wang 2006, pp. 42–49).

After the country was unified in the Sui dynasty (581–618), the five rites (*wuli* 五禮), i.e., auspicious rites (*jili* 吉禮), congratulatory rites (*jiali* 嘉禮), hosting rites (*binli* 賓禮), military rites (*junli* 軍禮), and inauspicious rites (*xiongli* 凶禮), were mainly inherited from

three sources. The first source was the rites from Liang梁 (502–557) and Chen 陳 (557–589) regimes, the second from Beiwei 北魏 (386–557) and Beiqi 北齊 (550–577) regimes, and the last one from Xiwei 西魏 (535–556) and Beizhou 北周 (557–581) regimes (Chen 2011, p. 3). There were three levels in the state ritual system of sacrifice. The top level, called grand sacrifice (*dasi* 大祀), was to offer sacrifice to the heaven, the earth, and others, followed by medium sacrifice (*zhongsi* 中祀) to *yue-zhen-hai-du* and others, and small sacrifice (*xiaosi* 小祀) to the stars, winds, rain, and others. Three places of offering sacrifices at the suburbs to the four seas in the Sui dynasty were related to the South Sea God. Firstly, at the Huangdici 皇地祠 (Shrine of the Earth God) which was 14 km north of the capital city Daxing 大興, the rulers worshiped their ancestors, during which the Jiuzhoushen 九州神 (Nine Precincts Spirit), seas, rivers, forests, ponds, hills, marshes, and terraces were all sacrificed to at the second grade simultaneously (Wei 1973, 6.108). Secondly, at the Yutan 雩壇 (Altar for Praying for Rain) which was 13 km south of the capital, *yue-zhen-hai-du* were sacrificed to at the second grade. As drought tended to occur after the fourth lunar month, a ceremony was performed at the altar for seven days to pray for rain which was believed to be brought by *yue-zhen-hai-du*. If no rain showed up, the ceremony would continue for another seven days conducted by officials and scholars who had made a contribution to the state. If the supplication was still unanswered, the third slot of seven days would be employed to pray for rain in the ancestral and imperial temples of the rulers. Again, if it still did not rain, the altar would be renovated to accommodate the ceremony for the fourth seven days. If the drought continued after all these endeavors, there was nothing the central government could do but repeat the cycle of sacrificing all over again. The local governments at provincial, prefecture, and county levels followed the same ritual as they prayed for rain towards the direction of the capital city's gates. If three rounds of sacrificing failed, they continued to pray to the sacred peaks, mountains, seas, and rivers. If the drought still continued, they prayed at the temples and shrines by offering bulls, goats, pigs as sacrifices (ibid., 7.128). Thirdly, at the Wujiaotan 五郊壇 (Five Suburbs Altar) a ceremony, called *zha* 蠟, was performed in the tenth lunar month to sacrifice to over a hundred spirits as a group. They included the gods of the sacred peaks, mountains, seas, and rivers, as well as the hills, forests, streams, and ponds. An additional spot was set along the one for the spirits of the sacred peaks, mountains, seas, and rivers. Therefore, a large number of spirits could be sacrificed simultaneously (ibid., 7.129–30). In this sense, the sacrifice rituals to *yue-zhen-hai-du* became a part of the suburban sacrificial institution at the capital during the Sui dynasty as the spirits were thought to be able to bring proper rain.

In addition to the suburban sacrificial ritual in the Sui dynasty, Donghaici 東海祠 (Shrine of the East Sea) was built by the coast of Kuaiji County 會稽縣 and Nanhaici 南海祠 (Shrine of the South Sea) was built in Nanhai town 南海鎮. Pines and cypresses were planted inside the shrines, and a priest called *wu* 巫 was appointed to maintain each shrine (ibid., 7.140; Wang 1960, 33.355). There are two reasons why the Shrine of the East Sea and the Shrine of the South Sea were singled out and set at the land of the previous regime Chen 陳, while the Shrine of Four Seas and the Shrine of the North Sea were not mentioned in the Sui dynasty. One reason is that such an arrangement was accorded with the geographical indication of the East Sea and the South Sea, and the other is that these two seas could defend the country as its territory expanded. The Shrine of the South Sea, therefore, is a perfect combination to indicate geography in a territorial and ceremonial way, and it remains the key venue enshrining the South Sea God for over 1400 years.

The Shrine of the South Sea erected in the Sui dynasty still stands at the Miaotou Village 廟頭村 in Huangpu District 埔區 in Guangzhou now, and its present name is the Nanhaishen Temple. Apart from building the shrine near the coast, the ancient rulers appointed priests in the neighborhood to clean the shrine, perform routine ceremonies, and adorn the yard by planting cypresses and pine trees which embodied the solemnity and reverence of the edifice. One year after the shrine was built, or in the third lunar month of the 15th year of Kaihuang (595) to be exact, Emperor Wendi "had imperial tours to the east and offered sacrifice to the five sacred peaks as well as the seas and waterways

at a distance" 至自東巡狩, 望祭五嶽海瀆 (Wei 1973, 2.40). About three months later, "a decree was passed to build shrines on famous mountains and great rivers that had not been sacrificed to yet" 詔名山大川未在祀典者, 悉祠之 (ibid.), which further extended the ritual scheme of sacrificing to *yue-zhen-hai-du*. In conclusion, a state ritual system of sacrifice to *yue-zhen-hai-du* was established in the Sui dynasty at the local level, in which the suburban sacrifice and the local sacrifice to the Shrine of the South Sea both played a role.

### 3. A Dual and Well-Established Scheme of Suburban Sacrifice and Local Sacrifice to the South Sea God in the Tang Dynasty

Unfortunately, the records of the rituals sacrificing to the South Sea God in the Sui dynasty were rare because the dynasty did not last long. In the Tang dynasty (618–907), the *Zhenguanli* 貞觀禮 (Rituals in the Reign of Zhenguan) and *Xianqingli* 顯慶禮 (Rituals in the Reign of Xianqing) were compiled. In particular, the *Datang Kaiyuanli* 大唐開元禮 (Kaiyuan Ritual of the Great Tang), which consists of 150 *juan*, was compiled by the academician Xiao Song 蕭嵩 and others and was completed in 732. According to this classic, the five rites were stipulated with 152 sub-rituals in total. Among the 55 sub-rituals of *jili*, offering sacrifices to the five sacred peaks and the four sacred strongholds ranked 47th, whereas to the four sacred seas and the four sacred waterways ranked 18th, hence these two sacrificial rituals were not the same (Du 1988, 106.2761–2762; Xiao 2000, 36.201–202). In fact, all the sacrificial rituals were distinct and strictly defined, as shown in the *Datang Kaiyuanli*, "Liyizhi" 禮儀志 (Records of Rites) in the *Jiu Tangshu* 舊唐書 (Old Tang History), "Liyuezhi" 禮樂志 (Records of Rites and Music) in the *Xin Tangshu* 新唐書 (New Tang History) and other records. For instance, as the medium sacrifice, praying to *yue-zhen-hai-du* shared the same ranking of praying to the state and the stars. It was lower ranking than the imperial praying to the heaven and the earth, which was the grand sacrifice, but higher-ranking than praying to winds and rain and to the general mountains, forests, rivers, and marshlands which belonged to the small sacrifice. The South Sea God was sacrificed to at the suburbs of the capital city Chang'an and the east capital Luoyang as one of the four sea gods at a second grade. During such suburban sacrificial rituals, including offering sacrifice to *diqi* 地祇 (earthly deities) on the summer solstice and to hundreds of spirits as a group on the eighth day of the twelfth month, the spirits of the four seas were sacrificed to at a second grade (Xiao 2000, 36.201–202; Du 1988, 106.2761–2762; Liu 1975, 24.911–912; Ouyang and Song 1975, 11.311–319).

When the Tang Emperors made an inspection tour, they offered sacrifices to Mount Tai 泰山 and also "sacrificed to mountain and water spirits at a distance arranged by a special sequence" 望秩於山川 as is recorded in "Huangdi Xunshou" 皇帝巡狩 (Inspection Tours of the Emperors) in the *Datang Kaiyuanli* (Du 1988, 118.3056–3060). The sacrificial ritual ranked from mountains, strongholds, seas, waterways, peaks, forests, rivers, marshes, plains, hills, and low meadows. Because the spirits of the mountains and waterways alike were thought to be able to bring proper clouds and rain, ceremonies were performed to pray for rain to come when there were droughts and for the rain to go when there were floods, and these ceremonies all involved sea spirits and others in *yue-zhen-hai-du*. For example, the sacrificial ritual to the sacred mountains and strongholds was conducted in the northern suburbs when there was a drought while serving *yue-zhen-hai-du* and all the mountain spirits at the same time. If the drought continued, sacrifices would be offered to pray for the state first and then pray at the imperial ancestral temple, followed by the ritual of praying to *yue-zhen-hai-du* (Liu 1975, 24.911–912; Du 1988, 120.3056–3060). Another example is that the sacrificial ritual would be performed at capital city gates or the state gates if there was a flood. If the supplication was unanswered, the same rituals would be performed as the one described above, "plus offering wine and dry meat" 並用酒脯 醯 (ibid.). In both cases, when the droughts and the floods ceased, ceremonies should be performed to thank and reward the spirits. However, the ceremonies coping with the natural disasters were only performed in the suburbs in an ad hoc manner, hence they were not on a par with the annual ritual in the local areas to sacrifice to *yue-zhen-hai-du*.

The sacrificial ritual to the South Sea God was rather complete in the Tang dynasty as the venue, and the dates were clearly stated. During the periods of Wude and Zhenguan, there was an annual sacrifice to the five sacred peaks, four strongholds, four seas, and four waterways, each of which took place on the day called "greeting the seasonal *qi* in the five directions" (*wufang ying qi* 五方迎氣)[2] that was based on the five-phase cosmology and matched the five quarters of the mountains and waters with the five seasons. In particular, the day of greeting the seasonal *qi* for the South Sea God was on the summer solstice. The spirits of the four seas were respectively sacrificed to at the shrines in Laizhou 萊州 by the East Sea, in Guangzhou by the South Sea, in Tongzhou 同州 (present-day east of Dali 大荔, Shaanxi) by distant sacrifice, and in Luozhou 洛州 (present-day Luoyang, Henan) by distant sacrifice. During the ceremonies, the imperial sacrifices (*taizai* 太宰), i.e., bulls, goats and pigs, were offered to the spirits, and the Supervisor (*dudu* 都督) and the Prefect (*cishi* 刺史) at the local government served as the chief supplicants (Du 1988, 46.1282). Although the gods of the West Sea and the North Sea were sacrificed to at the second grade elsewhere, the ritual scheme of mountain- and water-directed state sacrifices was already formed at the central government and fully implemented in the local areas. In summary, the state ritual system in the Tang dynasty was better designed than the ones in previous dynasties, because it perfectly denoted the four seas in both a political-cultural and geographical sense. When the ritual was performed to sacrifice to the South Sea God on the summer solstice, top-ranking government officials were appointed to officiate the ceremony. It demonstrates that the Tang government attached more importance to the sea gods than the Sui government, as the latter only appointed a priest to officiate the ceremony.

It is worthy of remark that during the period of Wude, prayer-board (*zhuban* 祝版) was used when sacrifices were offered to the spirits above the ritual rank of sacred mountain- and water-directed ones, and Mount Hua 華嶽 was sacrificed to by the emperor in person. After Empress Wu Zetian changed the name of her reign in 695, the sovereign was not supposed to sacrifice in person to the five sacred peaks, four strongholds, four seas, and four waterways according to the old state ritual. In other words, the sovereign could only offer sacrifices in name rather than in person. After several decades, the imperial court approved of a petition from the Court of Imperial Sacrifices (*taichangsi* 太常寺) in the first year of Kaiyuan (713) to change the old state ritual in which the Heir Apparent (*taizi* 太子) offered sacrifices to sacred mountain- and water-directed spirits by his name and by using prayer-boards. In accordance to the new ritual, "it was the emperor that sent a commissioner to offer sacrifices to sacred mountain and water spirits" 皇帝謹遣某乙, 敬祭於某嶽瀆之神 and wrote down his name in person for the supplication (Du 1988, 46.1283). In the first year of Shangyuan (760), the use of prayer-boards was forbidden when sacrifices were offered to the spirits below the rank of medium sacrifice such as *yue-zhen-hai-du* (ibid.). It was not until the fourth year of Zhenyuan (788), when the old ritual was resumed, that the use of prayer-boards offered by the emperors in person was re-introduced (Wang 1960, 33.369).

Medium sacrifice in the Tang dynasty was prepared by following the procedure of divination, abstinence, furnishing, cleaning, and displaying sacrificial vessels, paying homage, and burying (Ouyang and Song 1975, 11.311–319). To begin with, an auspicious date was carefully chosen as divined, preceded by three days of partial abstinence (*sanzhai* 散齋) in the residence and two days of complete abstinence (*zhizhai* 致齋) in the temple. In the course of partial abstinence, routine administrative affairs, except signing documents of judging crimes and executing punishment, were allowed to be attended to, but mourning for the dead, making inquiries about the sick, listening to music, eating meat, having sex, and anything related to the ritually polluting were abstained from. In the course of complete abstinence, nothing but performing sacrificial rituals was attended to. When the temple was furnished for the ceremony, things should be set or done in a certain direction and in a prescribed order. For instance, an altar should be set when sacrificing to the sacred mountains and strongholds, whereas a pit should be dug when sacrificing to the seas and waterways. On the altar, a monumental tablet should be put at the north while facing the south, whereas in the pit water should be filled and a roughly 3.3-m monumental tablet

should be set with steps in four directions. Subsequently, temples enshrining *yue-zhen-hai-du* were built with the statues of the spirits erected before the 9th year of the Zhenyuan (793) period. The old sacrificial rituals were all kept, such as setting up the altars and paying homage to the statues (Wang 2000, 8.786–788).

Vessels of sacrificing to the South Sea God differed over different periods in the Tang dynasty. For example, four bamboo-made vessels (*bian* 邊) and four wooden vessels (*dou*豆) were required in Wude and Zhenguan periods, while the number of both vessels was up to ten respectively at the ceremonies of medium sacrifice in the Xianqing period (Liu 1975, 24.911–912). Then in the Kaiyuan period (713–741), the sacrificial vessels for the five sacred peaks, four strongholds, four seas, and four waterways were specified as follows. Six bottles (*zun* 樽), ten bamboo-made vessels, ten wooden vessels, two round bowls (*gui* 簋), two square bowls (*fu* 簠), two big plates (*zu* 俎) were needed, together with bulls, goats, and pigs which were slaughtered and cooked. The wine was offered in the bottles, grain in the round bowls, and rice in the square bowls. On the bamboo-made vessels were salt, dried fish, dates, corns, hazelnuts, water chestnuts, starches, dried deer meat, white pastry, and black pastry; and on the wooden vessels were leeks, meat paste, *jin* pickles, deer meat paste, fish paste, *pi cai* pickles, and pork. On the day before the ceremony, the temple should be cleaned and furnished with the altar, monumental tablet, prayer-board and so on and so forth, and the spots for the chief, the second and the last supplicants and the hymn singer should be marked out. The process of performing the sacrificial ritual was rather lengthy. The chief supplicant began with washing and presenting a jade, followed by the priest who held the prayer-board and delivered the oration. The supplicant prayed and took a glass of wine from the priest who finished presenting the prayer-board on the altar, and the supplicant prayed again, bowing and kneeling, offered the wine, and then drank it himself. The priest showed up again with his subordinates to present the meat offerings and then passed them to the supplicant to pray for blessings. The second and the last supplicants followed the same procedure one after the other. The ceremony ended with all the vessels buried and the prayer-board burnt (Du 1988, 112.2897–2903; Xiao 2000, 36.201–202). From this specimen, it is apparent that the entire ceremony was grand and solemn.

In a nutshell, the sacrificial ritual system in the Sui and Tang dynasties demonstrates the imperial perception of "all under heaven" (*tianxia* 天下), as well as the imagination and definition of the territory. As Shinichiro puts it, "The ritual of sacrificing to the heaven at the round mound and to the earth at the square altar is part of the ritual of sacrificing to the heaven and the earth and even to the entire universe. The Son of Heaven, or the Emperors, employed the sacrificial ritual to prove the sanctioned legitimacy of the country" (Shinichiro 2008, p. 138). According to such a system of sacrificing to the spirits, the four seas were located at each end of the state under the heaven, instead of at a specified marine location. The four seas in the late Western Han dynasty, which was sacrificed to at the southern suburbs together with other spirits, should be interpreted as a conception that placed China in the center of the world; nevertheless, the four seas in the Sui and Tang dynasties, which were sacrificed to at the northern suburbs secondarily to other spirits, should be interpreted as a political-geographical conception. Beginning from the mid Tang, the role of the spirits changed, from a sanctioned political legitimacy for the country to embody different specified blessings for people to pray for. For instance, the titles of the four seas conferred by the emperors varied, as the East Sea God blessed people with proper winds and rain, the South Sea God with prosperous voyages and good harvests of fish and salt, the West Sea God and the North Sea God with abundant rain to stop droughts (Lu 2017, 6.65–67).

Unfortunately, no extant documents in the Sui dynasty record how the state system of sacrificing to *yue-zhen-hai-du* was implemented in the local areas. In contrast, the records in the Tang dynasty offered a paradigm for the coming generations to follow when sacrificing to *yue-zhen-hai-du*. Moreover, the sacrifices to the five sacred peaks, four strongholds, four seas, and four waterways were ranked as the medium sacrifice of the imperial court,

so the rituals performed by the local government were entrusted by the imperial court and became the top-ranking sacrificial ceremony in the local areas. As far as the South Sea God was concerned, the emperors sent commissioners to Guangzhou to officiate the ritual ceremonies.

#### 4. The Two Brothers Zhang Jiuling and Zhang Jiuzhang Were Appointed as Ritual Commissioners to the South Sea God during the Reign of Emperor Xuanzong

The Zhangs were the most renowned family in Lingnan as the three brothers Zhang Jiuling 張九齡, Zhang Jiugao 張九皋 and Zhang Jiuzhang 張九章 were all high-ranking officials in the imperial court in the Tang dynasty. Zhang Jiuling was the Secretariat Director (*zhongshu ling* 中令), Zhang Jiugao was the Director of the Palace Administration (*dianzhong jian* 殿中), and Zhang Jiuzhang was the Minister of the Court of Imperial Entertainment (*honglusi qing* 寺卿) (Liu 1975, 99.3098–3099; Ouyang and Song 1975, 126.4428). In addition, the Zhang family also took office in the local government in the Lingnan region. Moreover, Zhang Jiuling and Zhang Jiuzhang were both appointed to Guangzhou as commissioners to perform the sacrifice to the South Sea God.

Zhang Jiuling, who was then the Vice Minister of the Court of Imperial Sacrifices (*taichang shaoqing* 太常少卿), was sent to sacrifice to the South Mountain and the South Sea in the 14th year of the Kaiyuan period (726) as the country suffered from severe droughts (Wang 1960, 144.1752). As the name shows, the Court of Imperial Sacrifices was in charge of the sacrificial rituals to all the deities in the country, and the Vice Minister served as the aide with the fourth upper official rank, or rank 4a. He had been demoted to this rank due to his relationship with Minister Zhang Yue 張 who had been deposed (Liu 1975, 99.3098). He made his way in the sixth month of the year to complete the imperial mission, and then he visited his hometown. We can follow his footsteps in all the poems he wrote along the way, which were published in the third and the fourth *juan* of *Qujiangji* 曲江集 (Qujiang Anthology). The titles of his poems are as follows: "Ascending the Mount Yu in Lantian County from where I went South as a Commissioner" 奉使自藍田玉山南行 (Zhang 1986, pp. 183–84), "On My Way to the South Sea as a Commissioner on a Summer Day" 夏日奉使南海在道中作 (ibid., 185–86), "Heading to the South from the Xiang River" 自湘水南行 (ibid., 13), "Visiting Sima the Taoist Priest after Ascending the South Mountain" 登南嶽事畢謁司馬道士 (ibid., 195–96), and "Arrival at Guangzhou as a Commissioner" 使至廣州 (ibid., 270). Judging from the titles, we can conclude that he set out to the Southeast from Chang'an by way of Lantian, Xiangzhou 襄州 (present-day Xiangyang 襄陽, Hubei) and Jingzhou 荊州. After he reached Yuezhou (present-day Yueyang 岳陽, Hunan) along the Yangtze River 長江, he continued following the Xiang River 湘江 to Hengzhou 衡州 (present-day Hengyang 衡陽, Hunan). Then he arrived at Mount Heng 衡山 to offer sacrifices and headed south via the Qitian Mountain 騎田嶺 and the Gorge Zhenyang 湞陽峽 to his destination, Guangzhou. He described in one of the poems that "I travel over ten thousand *li* on the hottest days in midsummer" 緬然萬里路, 赫曦三伏時 (ibid., 185). When sacrificing to the South Sea God in Guangzhou, he openly admitted that "I finish my job with reverence and now I can attend to my personal matters" 肅事誠在公, 拜慶遂及私 (ibid., 185). As a matter of fact, he went back to his hometown to visit his family after the business, and then he returned to the North via the Dayu Mountain 大庾嶺 and the Gan River 贛江.

Unlike the eldest brother Zhang Jiuling who was sent to pray for ending the droughts, another Zhang brother was sent to sacrifice to the South Sea in the 10th year of Tianbao (751) for a different reason: to confer titles to the four seas on behalf of the emperor to acknowledge the divine standing of the spirits. The status of the sea spirits was raised together with the ones of the five sacred mountains and four waterways during the reign of Emperor Xuanzong. In the 5th year of Tianbao (746), for instance, the emperor bestowed titles on all the five sacred mountains. He continued to confer the title of Duke (*gong* 公) to all the four sacred waterways in the next year (747) and the title of King (*wang* 王) to the four seas in the first month of 751. Interestingly, different names were conferred with the titles upon the four seas for special connotations: the East Sea God, Guangdewang 廣王

(King Guangde), meaning to teach good morals broadly; the South Sea God, Guangliwang, meaning to generate wealth massively; the West Sea God, Guangrunwang 廣潤王 (King Guangrun), and the North Sea God, Guangzewang 廣澤王 (King Guangze), both meaning to grant proper rain and waters. It is apparent that all these four names were related to the cultural and geographical situations in the local areas. For instance, the name "Guangli" was chosen because of the fact that Guangzhou could import a large number of foreign treasures by trade (Wang 2006, p. 67). Moreover, the titles of the spirits of the four seas were equal to the ones of the five sacred mountains, and were higher-ranking than the ones of the waterways.

There are conflicting records in the historical archives concerning who was sent to sacrifice to the South Sea God by the emperor in 751. I argue it is Zhang Jiuzhang who was the imperial commissioner based on the following records. In "Si Yuezhenghaidu" 祭嶽鎮海瀆 (Sacrificing to *Yue-zhen-hai-du*) in the *Datang Jiaosi Lu*, a correction is made to identify Zhang Jiuzhang, rather than Zhang Jiugao, as the commissioner (Wang 2000, 8.786–788). In other records, including "Liyizhi" in the *Jiu Tangshu* (Liu 1975, 24.934); "Chong Jisi" 崇祭祀 (Sacrificial Rituals), "Diwang Bu" 帝王部 (Section of the Emperors) in the *Cefu Yuangui* (Wang 1960, 33.365); "Fuzhai Beilu" 復齋碑 (Stele Inscriptions of Fuzhai) in the *Baoke Congbian* 寶刻叢編 (Anthology of the Inscriptions in the Song Dynasty) (Chen 2012, 19.1113); and "Ceji Guangliwang Ji" 冊祭廣利王記 (Records of Sacrificing to King Guangli) in the *Quan Tangwen* 全唐文 (Complete Prose Works of the Tang Dynasty) (Dong 1983, 987.1023), we find the same statement that Zhang Jiuzhang was sent to officiate the ceremony. In addition to these records, I have another three points of justification for my argument.

Firstly, the official rank of the commissioner. The three brothers finished mourning for their dead mother in the sixth month in 736, and then Zhang Jiuling recorded that "one of my younger brothers Jiugao was appointed to be the Palace Administrator (*dianzhongcheng* 殿中丞) while the other one Jiuzhang was Court Gentleman for Consultation of the Heir Apparent (*taizi siyilang* 太子司議郎) "(Zhang 1986, p. 578). As far as the official rank was concerned, Zhang Jiugao enjoyed a higher place than his younger brother Zhang Jiuzhang because he held the 5b1 rank while his younger brother held the 6a1 rank. We can also find all the ranks he held at different positions throughout his life in his epitaph, including the 5b1 rank as Director of the Department of State Affairs (*shangshu zhifang langzhong* 尚書職方郎中), the 4a2 rank as Governor (*junshou* 郡守) of Ankang 安康, an the 3b rank as Governor of Huai'an 淮安, Pengcheng 彭城, and Suiyang 睢陽 respectively (Li 1966, 899.4731–4733). However, none of them matches the 4b1 rank of Aide of the Princely Establishment (*wangfu zhangshi* 王府長史) of the commissioner who was sent to Guangzhou.

Secondly, the poem titled "Farewell to Zhang Sima of the Hanlin Imperial Academy on the Way to the South Sea" 送翰林張司馬南海勒碑 (Huang and Huang 1987, 19.735). Liang Quandao 梁權道, the Song scholar who edited the poems of Du Fu 杜甫, stated that the poem was written in the first year of Qianyuan (758) period, but Huang Xi 希 and Huang He 鶴, two scholars who added footnotes to the poems, found that there was no such a position called Commander (*sima* 司馬) in the Hanlin Imperial Academy when they checked "Baiguan Zhi" 百官志 (Record of Hundreds of Government Officials) in the *Xin Tangshu*, though there was a Commander in the suite of the commissioner who was sent by the emperor to sacrifice to the South Sea. They, therefore, believed that Zhang worked in the Academy without the title of Commander (ibid.). Judging from the life experience of Du Fu (712–770), I agree with Huang and Huang that the poem was not written in 758, but in 751 when Du Fu was at the Academy in the capital city. The poet probably got confused with the commissioner's official rank as the Aide of the Princely Establishment and the Commander, both of which belonged to the fourth rank, but the former was still higher than the latter. It is also possible that Zhang Jiuzhang was just new to his position as the Aide of the Princely Establishment, which was not known to the poet yet, as we can find evidence in the epitaph, currently kept by the library of Luoyang Normal University, of his eldest son. The son was called Zhang Zhao 張招, and he passed away in 749 when

his father was still Aide of the Princely Establishment ([Guo and Yang 2015](), pp. 32–33). Therefore, Du Fu's poem was dedicated to Zhang Jiuzhang.

Thirdly, the position of the commissioner at the local government. In "Ceji Guangliwang Ji", we read "commissioner Zhang is the magistrate of Nanhai previously" 初, 張公作宰南海 ([Dong 1983](), 987.1023). As a matter of fact, Zhang Jiuzhang served as the magistrate of Nanhai County for a period of time, but Zhang Jiugao never did, though the latter served as the Prefect of Guangzhou and the Military Commissioner of Lingnan (*Lingnan jiedushi* 嶺南節度使) from 751 to 753 ([Yu 2000](), p. 3163). The two brothers both took part in the sacrificial ceremonies, but they represented different positions: one was an imperial commissioner and the other was a local government official, and the former one was undoubtedly Zhang Jiuzhang. The confusion is partly caused by the local chronicles in Guangdong as well as by *Boluo Waiji* 波羅外紀 (Stories of Boluo Temple) written by Cui Bi 崔弼 in the Qing dynasty. In fact, the stele inscriptions in the temple were all lost in 751, and, thus, Cui Bi mistakenly recorded Zhang Jiugao as the commissioner ([Cui 2017](), 6.92).

## 5. Differences of the Sacrificial Rituals to the South Sea God in the Early and the Late Tang Dynasty

Since the Nanhaishen Temple was far from the capital city in the Tang dynasty, the local government officials were, therefore, usually in charge of sacrificing to the deity. It is worth mentioning that in the early Tang, or before the An Lushan Rebellion (755–763), an imperial commissioner was usually sent to Guangzhou to officiate the ceremonies, demonstrating the implementation of state ritual system in the local areas. In the late Tang, however, local government officials usually sent their deputies to officiate the ceremonies due to the declining national power and social instability, and other problems in the Lingnan region.

Although it became a new norm for the local government officials to replace the imperial commissioners to officiate the sacrificial ceremonies to the South Sea God, the emperor also sent his commissioners to Guangzhou from time to time due to natural disasters and cultural reasons. Apart from the two Zhang brothers, a couple of other imperial commissioners were also sent to sacrifice to the South Sea God in the Tang dynasty ([Wang 2006](), pp. 462–68).

Emperor Xuanzong longed to be immortal, and therefore bestowed titles to the five sacred mountains in the Kaiyuan period and to the four sacred waterways and four sacred seas in the Tianbao period. As mentioned above, "the four seas were given the titles of King by the emperor" 四海並封為王 in the first month of 751, and Zhang Jiuzhang was dispatched to Guangzhou to confer the South Sea with the title "Guangliwang" on behalf of the emperor ([Wang 1960](), 33.365).

Emperor Xuanzong issued as many as 23 decrees to perform the rituals of a mountain- and water-directed state sacrifices in Kaiyuan and Tianbao periods. As far as the South Sea God was concerned, the Prefect of Guangzhou officiated the annual sacrificial ceremony on the summer solstice.

In addition to the annual ceremonies, the South Sea God was sacrificed to on an ad hoc basis at four occasions, as the *Cefu Yuangui* indicates as follows.

Firstly, at the occasion of praying to *yue-zhen-hai-du* for proper rain, particularly during the reign of Emperor Xuanzong. For instance, in the first month of 730, the fourth month of 731, the fourth month and eleventh month of 732, the first month of 735, the first month of 747, the sixth month of 749, the first month of 751, and the second month of 753, sacrificial rituals were performed. Among these eight rituals, two were to confer titles to the four waterways in 747 and to the four seas in 751 respectively, while the rest were related to the emperor himself who was so "concerned with the myth of immortality" 尚長生輕舉 之術 that he attached great importance to sacrificing to the spirits ([Liu 1975](), 24.934). His successors also ordered the rituals to be conducted, namely in the second month of 764, the sixth month of 770, the fourth month of 786, the first month of 807 and the sixth month of 827. At these five occasions, as well as the eight mentioned above, local government officials were usually the main supplicants to the water and mountain spirits which could

bring proper winds and rain, while imperial commissioners were sent to officiate from time to time (i.e., in the years of 726, 731, 751, 770 and 786) (Wang 1960, 34.367–369).

Secondly, at the occasion of praying for ending the droughts and rewarding the deity for good harvests. Fifteen ceremonies were performed to pray for ending the droughts, which were prone to occur in spring and summer, including in the sixth month of 630, the second month of 669, the first month of 706, the fifth month of 715, the fifth month of 721, the sixth month of 726, the sixth month of 728, the tenth month of 737, the ninth month of 749, the second month of 751, the eighth month of 755, the third month of 759, the sixth month of 767, the third month of 790, and the seventh month of 803. As a result, *yue-zhen-hai-du* and other water and mountain spirits were sacrificed to for proper rain (ibid., 144.1764–1757). The local government officials were the main supplicants for these ceremonies, except the ones in 726 and 790 when imperial commissioners were sent to the local shrines. When the supplications were answered with good harvests or proper rain, the spirits were rewarded with gratitude at the ceremonies, such as the ones in the sixth month of 728, the sixth month of 734, the tenth month of 737, the twelfth month of 741, the fourth month of 744, the ninth month of 749, and the eighth month of 755 (ibid., 33.359–366).

Thirdly, at the occasion of the emperors taking the throne and changing their holy titles and the names of their reigns. Examples can be found in the fifth month of 748 and the seventh month of 821 when collective petitions were made by the imperial officials to suggest the emperors rename their holy titles, and *yue-zhen-hai-du* spirits were sacrificed to after the emperors approved of the petitions (ibid., 33.364, 34.364–369). Moreover, when the emperors changed the names of their reigns, such as in the first month of 724, the fourth month of 760, the first month of 765, the eleventh month of 766, and the fourth month of 785, *yue-zhen-hai-du* and other water and mountain spirits were sacrificed to (ibid., 33.361, 34.367–368). It is worth pointing out that Emperor Wenzong decreed to reward the five sacred mountains and four waterways and others by offering them sacrifices in the second month of 834 because he recovered from a disease. He celebrated his recovery by proclaiming a general amnesty and ordering the top officials at the local governments to offer thanksgiving sacrifices to the water and mountain spirits that had blessed him with good health (ibid., 34.369).

Fourth, at the occasion of conferring titles to the Heir Apparent. Examples can be found in the fourth month of 805 for Li Chun 李純 (who later became Emperor Xianzong) to be canonized, in the tenth month of 809 for Li Ning 李寧 (who died young), in the tenth month of 812 for Li Heng 李恒 (who later became Emperor Muzong), and so on. Prefects at local areas were assigned to officiate these ceremonies to inform and sacrifice to the water and mountain spirits (ibid.).

To summarize, state rituals of sacrificing to *yue-zhen-hai-du*, including the South Sea God, were conducted as the emperors made every attempt to maintain their supreme rules, particularly, when there were droughts, emperors changing the names of their reigns and their holy titles, and designating their successors. *Yue-zhen-hai-du* embodied the jurisdictional right of the country in the geographical and political-cultural sense. It is, nevertheless, necessary for us to demonstrate how the local government officials implemented the state sacrificial rituals.

It was a risky trip for the government officials to attend the annual ceremony at the Nanhaishen Temple on the summer solstice. They had to travel 80 *li* (i.e., 36 km) on a bobbing boat to the east of the city, which was then frequented by monsoons and typhoons, thus, their boats could be easily blown over, and they risked their lives as they were heading against the violent storms and the roaring waves. In the early Tang when the country was at its prime, the top official in Guangzhou was dispatched to be the chief supplicant to the South Sea God. In the late Tang when the country was waning and torn by warlords, however, the state ritual of sacrifice was often barely performed. The local officials in the late Tang were so scared of the risky boat trip that they declined to go either by lying that they were sick (Han 1986, 31.485–489), or they simply sent their deputies or assistants to

the ceremonies on their behalf (Liu 1975, 154.4098). But there was an exception. Kong Kui 孔戣, who took office as the Prefect of Guangzhou and Military Commissioner of Lingnan in the seventh month of 817, was determined to make his way to the temple on the day before the annual ceremony in 818 regardless of the stormy weather and the obstruction of his subordinates. Fortunately, he arrived safe and sound and spent the night there. When he woke up, the weather turned out to be fine, and, thus, the ceremony was held as grand as it should be. He and his colleagues all put on their best official robes to stand in lines, the sacrificial vessels were all clean and tidy, the offerings were all set in a good order, and the ritual music was echoed at the bustling temple. Interestingly, the rest of the year witnessed no more storms but an excellent harvest. Kong Kui continued officiating the ceremony in person the next year and ordered the temple to be enlarged and renovated. Again, for the third year in a row, he went with his colleagues to sacrifice to the South Sea God, which subsequently did bless the region with a good harvest. All his endeavors were fully described by Han Yu 韓愈, a famous contemporary writer, in his essay titled "Stele Inscription of the Temple of the South Sea God (Guangliwang)" 南海神(廣利王)廟碑 (Han 1986, 31.485–489).

The successors of Kong Kui did not sacrifice to the South Sea God as regularly as he did. Yet we can still find examples of the top officials in Guangzhou to officiate the ceremonies, such as Li Pin 李玭, who was in office from 847 to 848 (Wu 1980, p. 1036). It was recorded in the poem titled "Poem on Clan Uncle Lianggong's Spring Sacrifice to the Temple of King Guangli" 涼公從叔春祭廣利王廟詩 written by Li Qunyu 李群玉 (Li 1987, p. 49). It is worth pointing out that this local ritual of sacrifice in spring shared the same goal as the imperial one on the summer solstice: to pray for a good harvest and peace of the dominion. Another example can be found in the fourth month of 864 when a rebellion broke out in the Lingnan region. Gao Pian 高駢 was hence appointed to offer amnesty and enlistment to rebels. Before setting off, he visited the shrine to pray for a safe voyage as he wrote in his work titled "The Temple of the South Sea God" 南海神祠 (Gao 1980, 598.6918). Apparently, the deity played an important role in maritime transportation in this case. To summarize, the South Sea God had increasing visibility in Lingnan, and the sacrificing to it in the late Tang was mainly related to its blessings for no disastrous storms, no poor harvests, and no social instability.

## 6. Reciprocity between the South Sea God Belief and Buddhism with the Establishment of Linghua Monastery to Calm Down the Stormy Sea

The bay by which the Nanhaishen Temple is built is rather turbulent. Located at the crossing of the sea and the Pearl River estuary, the funnel-shaped waterway is so narrow and long that boats are prone to be blown over by the strong winds and waves. As a result, people in ancient times blamed the hot-tempered deity that drowned many people passing by (Jiang 2007, 20.144). As it was the biggest religious event in Lingnan to sacrifice to the South Sea God in the Tang dynasty, some Buddhists appeared to take advantage of the so-called hot temper of the sea god to build a temple nearby by telling the legend that the god was converted to Buddhism.

In the first month of the second year of Yuanyou (1087) in the Northern Song dynasty, the Prefect of Guangzhou, Jiang Zhiqi 蔣之奇, paid homage to the Nanhaishen Temple. After that, he visited Linghua Monastery 靈化寺 and learned about the legendary relationship between Buddhist Master Xiujiu 休咎禪師 and the South Sea God by obtaining the ancient stele of Master Daoheng 道行大師 of the temple. Legend has it that, from the 6th year (790) to the 8th year of Zhenyuan (792),[3] Li Fu 李復, the Prefect of Guangzhou and Military Commissioner of Lingnan, once sent a soldier called Li Yu 李玉 from Luofu Mountain 羅浮山 to Fuxu Town 扶胥鎮 on the southeast to welcome Buddhist Master Xiujiu. As they spent the night in the western chamber of the temple (which was then also known as Zhenhai General's Temple 鎮海將軍廟), two young boys in a green dress came at midnight and asked the Buddhist master, "Why do you come here? Don't you know the mighty power of the South Sea God?" A couple of hours later, there came a thunderstorm and the

South Sea God who was dressed in a purple and gold robe appeared. The master then confronted the deity and asked him to turn the temple into a Buddhist one, but his request was declined. Nevertheless, the deity offered the master another slot to build the Buddhist temple, which was subsequently turned into Linghua Monastery. The master learned that a large number of people had been drowned in front of the temple, and he, thus, wanted to save people's life by converting the hot-tempered deity into a gentle Buddhist. After he was converted, the South Sea God followed the Dharma to calm down the winds and waves on the sea, which encouraged people to believe in Buddhism as the mighty deity did (ibid., 144–45).

The legend that Master Xiujiu converted the South Sea God into a Buddhist was recorded in the Song dynasty. For example, in the *Yudi Jisheng* 輿地紀勝 (Geographical Record) written by Wang Xiangzhi 王象之, we can find the same legend (Wang 1992, 97.3051–3052). Moreover, a famous general called Li Gang 李綱 wrote a poem titled "Visit to the Temple of the South Sea God" 謁南海神廟 as he passed by the Nanhaishen Temple in Guangzhou in the third year of Jianyan (1129) and recorded the same legend as well (Li 2004, 26.344). And the legend continued to be recorded in other documents such as "Linghuasi" 靈化寺 (Poem on Linghua Monastery) written by Fang Xinru 方信孺 to give credit to the master who converted the deity to stop the storms on the sea (Fang 2010, p. 38).

The above legend shows that Buddhists used the dialogue between the master and the deity to convert the latter as the former's disciple, which was a way to promote Buddhism. As a matter of fact, it is the Buddhists that took advantage of the South Sea God to promote the Dharma. On the one hand, the South Sea God was equal to the five sacred mountains spirits, and, thus, it enjoyed a great reputation in Lingnan. Buddhists took advantage of this reputation to empower their own religion. Since the master wanted to occupy the temple, it was in his interest to accept the deity as a Buddhist disciple whose temple would therefore become a Buddhist monastery. With the help of this legend, it was the second-best solution to build a Buddhist monastery near the deity. On the other hand, it shows that the South Sea God worshiped by the state was able to benefit from Buddhism, in having his "hot-tempered" reputation transformed by means of this new legend. After converting to Buddhism, the deity became so kind and docile that he stopped the turbulence on the sea. Moreover, there was no conflict but reciprocity between the deity and the master as they both had their own temples, particularly the latter who could use the legend to establish the Linghua Monastery in a legitimate way. As it is, the name of Linghua means "Numinous Transformation" of Buddhism and, thus, the temple became on a par with the Nanhaishen Temple.

It is worth mentioning that in the twenty-ninth year of Kaiyuan (741) Buddhist master Bukong 不空 visited Guangzhou for the second time when he was on his way to the Lion nation (present-day Sri Lanka). The Investigation Commissioner of Lingnan (*lingnan caifangshi* 嶺南採訪使) Liu Julin 劉巨麟 asked the master for Abhiṣeka, an enlightenment ceremony, to a large number of people at Faxing Monastery 法性寺 (present-day Guangxiao Monastery 光孝寺), which thus became a big event in the local area (Zan 1987, 1.7–8). Years later in Zhenyuan period, Li Fu invited Master Xiujiu to Guangzhou again. The two officials had the same reason for their invitation to the Buddhist masters: to set a local religious order and to maintain social stability.

## 7. Systematization of Ritual Sacrifice to *Yue-Zhen-Hai-Du*

In the early Tang, no state rituals were prescribed and, thus, the ritual was discussed ad hoc according to the *Zhenguanli* (which was composed of 138 articles in total, including 60 articles in the *Jili*, 4 in *Binli*, 20 in *Junli*, 42 in *Jiali*, and 11 in *Xiongli*) and to the *Xianqingli* (which was composed of 130 *juan*). All the practices of holding ritual ceremonies and sending ritual commissioners laid a theoretical foundation and offered case studies to the *Datang Kaiyuanli.* In the period of Kaiyuan, the ruler approved of the petition made by Zhang Yue to stipulate the state rituals by learning from the previous rituals in the periods of Zhenguan and Xianqing. He suggested that the state rituals should compromise all the

similarities and differences to re-edit *Liji* 禮記 (Record of Rituals). The 20th year of Kaiyuan (732) period witnessed the issuing of the *Datang Kaiyuanli,* which was initiated by Xu Jian 徐堅 and completed by Xiao Song and others. The 150-*juan* classic thus became the paradigm of the ritual system which included all the five rituals. People in later generations followed it with slight modification, as it was so well-established that no other works could surpass it (Ouyang and Song 1975, 11.309). The Tang scholar Du You 杜佑 took an excerpt from the classic into a new work titled *Tongdian* 通典 (General Institutions), and a similar approach was adopted in the compilation of the *Jiu Tangshu* and *Xin Tangshu* with footnotes and annotations. In the 9th year of Zhenyuan (793), Wang Jing 王涇 compiled the *Datang Jiaosi Lu*, also known as the *Tang Zhenyuan Jiaosi Lu* 唐貞元郊祀 (Suburban Sacrifices in Zhenyuan of Tang), which recorded the state ritual system of sacrificing to the spirits at the suburbs in the Tang dynasty based on the *Datang Kaiyuanli* which was conclusive, comprehensive, and systematic (Zhao 1994, pp. 87–91). With all the new texts in place, therefore, the five rituals in the early Tang became more prescriptive which subsequently led to the framework of state rituals in the late Tang. Such state rituals in the Tang dynasty, purposefully distinct from the ones in previous dynasties, embodied the power and prosperity of the country, the pursuit, and innovation of the emerging bureaucrats and scholars, the adaptation to the needs of the times, the authority of the emperors, and the function of guiding the politics at court and the social life of people. In short, as a superstructure ritual, it was synchronized with the development of the society and the economy, and its formation was highly purposeful and pragmatic (Wu 2005, pp. 73–94).

Subsequently, classics such as the *Kaiyuan Houli* 開元後禮 (Rituals after Kaiyuan) and *Qutai Xinli* 曲臺新禮 (New Rituals of Qutai) in the late Tang, *Taichang Yingeli* 太常因革禮 (Rituals in the Northern Song Dynasty) (Ouyang and Su 2002) and *Zhenghe Wuli Xinyi* 政和五禮新儀 (Five New Rituals of Zhenghe) (Zheng 1987) in the Northern Song dynasty, *Dajin Jili* 大金集禮 (Collection of Rituals in Jin) (Zhang 1985) in the Jin dynasty, and *Qinding Daqing Tongli* (1987) 欽定大清通禮 (Imperial Approved Rituals in the Great Qing), all inherited the rituals prescribed in the *Datang Kaiyuanli* whose scale and influence was still insurmountable. It is not only the paradigm of the ancient rituals in China but also a role model in East Asia with a significant impact on the local ritual system and legal regulations. According to Ikeda, the state ritual systems in Balhae, Silla, Japan, and Goryo all learned from the *Datang Kaiyuanli,* particularly Japan that copied the entire ritual system of the Tang dynasty. Moreover, the Tang classic offered abundant cases of decrees and laws, which are rare and precious historical materials in the legal history of the dynasty. Among them, the disputes about the classics and rituals are important materials for the study of the history of thoughts and classics in the Middle Ages. As Ikeda summarizes, the *Datang Kaiyuanli* provides a large number of data and a new perspective for scholars in the fields of history, anthropology, and culture (Ikeda 1992, pp. 165–93).

The same was true of the sacrifice to *yue-zhen-hai-du*. The dual scheme of suburban sacrifice and local sacrifice was implemented beginning from the Kaiyuan period. Though the places of the sacrifices to the East Sea, the North Sea, and the West Sea changed constantly, the one-off sacrifice to the South Sea never changed throughout the various dynasties. It was the only permanent venue among the ones of sacrificing to the four seas because it had always been within the territory ruled by the different emperors (except by the emperors of the Jin dynasty and of the Five Dynasties who failed to rule beyond the south to Qinling 秦嶺 and Huai River 淮河), which resulted in a lot of stele inscriptions and relics well preserved to the present. The rituals of suburban and local sacrifices to *yue-zhen-hai-du* were also preserved as they were the role models for people to look up to when worshiping the South Sea God, such as the tradition of sending a commissioner and assigning a top local official to officiate the ceremonies. The deity was worshiped commonly both by the imperial court and by the local people in general as it became increasingly popular in Lingnan (Wang 2006, pp. 98–444).

### 8. Conclusions

In this article, I have studied the state ritual system of sacrificing to *yue-zhen-hai-du* in general and sending commissioners to officiate the ceremonies worshiping the South Sea God in particular. My aim has been to reveal how such a well-established scheme was integrated with the implementation of the *Datang Kaiyuanli,* and how such a state ritual policy was carried out in the local areas with the Nanhaishen Temple as a case study. I conclude that the sacrificial ceremonies for the deity go beyond a mere form of official sacrifice and demonstrate the national geography in the "all under the heaven" sense, as well as the state political and cultural power in Lingnan. The state kept the religious right to sacrifice to the South Sea God, no matter whether it was the imperial commissioners or the local government officials that were sent, as they both officiated the ceremonies on behalf of the state.

Firstly, the state suburban sacrifice to *yue-zhen-hai-du* was implemented in the local areas in the Sui dynasty and completed as a whole in the Tang dynasty. As far as the South Sea God was concerned, there was a change from a suburban sacrifice in the early Tang to a dual scheme of suburban and local sacrifices in the late Tang, which belonged to the medium sacrifice. On every summer solstice, the annual ceremony to worship the South Sea God was performed with the procedure of assigning prayer-board, abstinence, displaying sacrificial vessels, checking the vessels of presenting the three animal sacrifices, identifying the spots of the supplicants, praying to the deity with three rounds, playing the music, and burying all the sacrificial items. Every step at the ceremonies was strictly prescribed, which led to a well-established ritual scheme of sacrifice.

Secondly, the Zhang brothers who were famous in the Lingnan region were sent by Emperor Xuanzong to sacrifice to the South Sea God. In the sixth month of 726, Zhang Jiuling was sent to sacrifice to the South Mountain and the South Sea due to severe droughts in the country. After he finished this imperial mission, he visited his hometown in Guangdong. Unlike his eldest brother, Zhang Jiuzhang was sent in 751 to worship the four seas, including the South Sea, in order to confer titles to the spirits of the four seas and show the emperor's reverence. There are confusing statements in historical records concerning whether it is Zhang Jiugao or Zhang Jiuzhang who was sent as the commissioner in 751, and I argue that the commissioner should be Zhang Jiuzhang as evidenced in the official titles, the epitaphs, and other records.

Thirdly, the officials who worshiped the South Sea God as the chief supplicant in the Tang dynasty were mainly the top local officials, i.e., the Prefect of Guangzhou. In the early Tang, the well-established state rituals of sacrifices were carried out effectively. In the late Tang, however, the officials were so scared of the turbulent winds and waves on the sea that they sent their deputies to attend the ceremonies. But there were exceptions, such as Kong Kui, Li Pin, and Gao Pian, who worshiped the deity in person. In the late Tang, the local officials offered sacrifices to the deity mainly for ending disastrous storms, poor harvests, and social instability. As the state ritual system of sacrifice was gradually carried out in Lingnan, the role of the South Sea God became more visible than ever before.

Fourth, there was reciprocity between the South Sea God belief and Buddhism in the Tang dynasty. Legend has it that Master Xiujiu converted the deity to turn the temple into a Buddhist monastery as the deity partially agreed by allotting another slot in the neighborhood to build the Linghua Monastery. In return, the deity, who was a Buddhist disciple then, had a better reputation for calming down the turbulent sea and thus stopping drowning people on their voyages. We can learn from the legend that Buddhism and the state ritual system of worshiping the deity had a reciprocal agreement as they could both bless people with safe voyages and social stability.

Lastly, it became usual in the Tang dynasty to conduct both suburban sacrifices to *yue-zhen-hai-du* as secondary to the main deities and local sacrifices to *yue-zhen-hai-du* as the main deities themselves. Moreover, the *Datang Kaiyuanli* improved the state ritual system, particularly the *jili*, and laid a solid foundation for the late Tang and the following dynasties of Song, Jin, Yuan, Ming, and Qing. As a matter of fact, the classic not only

became a paradigm for the future generations to sacrifice to the South Sea God but also exerted influence on the sacrifice to *yue-zhen-hai-du* for more than one thousand years. Moreover, the changes of the sacrificial rituals to the South Sea God throughout dynasties reflect how the state ritual system of sacrificing to *yue-zhen-hai-du* was implemented at the local level. In the end, as it was fully popularized and localized in Lingnan, the South Sea God was jointly worshiped by the government and by the general public.

**Funding:** National Social Science Fund of China: 15AZS009.

**Institutional Review Board Statement:** Not applicable.

**Informed Consent Statement:** Not applicable.

**Data Availability Statement:** Not applicable.

**Conflicts of Interest:** The author declares no conflict of interest.

## Notes

[1] The term *yue-zhen-hai-du* 嶽鎮海瀆 refers to the five sacred peaks (*wuyue* 五嶽), five strongholds (*wuzhen* 五鎮), four seas (*sihai* 四海), and four waterways (*sidu* 四瀆) in a group, instead of the five sacred peaks and four waterways only. This term is used in the records of the ritual system in the Tang and other dynasties, such as the *Xin Tangshu* 新唐書 (New Tang History), *Jiu Tangshu* 舊唐書 (Old Tang History), *Datang Jiaosi Lu* 大唐郊祀 (Records on the Suburban Sacrificial Rituals in Tang), *Cefu Yuangui* 冊府元龜 (Song Dynasty Historical Encyclopedia) and so on. The term, therefore, is used by the author in this paper as well unless otherwise stated.

[2] For a detailed discussion of this ritual, see Niu (2017, pp. 105–12).

[3] Jiang Zhiqi made a mistake in recording Li Fu as the Prefect of Guangzhou and Military Commissioner of Lingnan in the tenth year of Tianbao (753) in his works "Linghuasi Ji" 靈化寺記 (A Record of Linghua Monastery). Li Fu took his position of Military Commissioner of Lingnan in 790–792. See Yu (Yu 2000, p. 3168). Moreover, by comparing the life experiences of Li Fu (739–797) and Master Xiujiu (746–807), we can conclude that the former should entertain the latter during the Zhenyuan period instead of the Tianbao period.

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
