# Peer review of "The Sacrificial Ritual and Commissioners to the South Sea God in Tang China"

_religions, doi:10.3390/rel12110960_

Round 1

Reviewer 1 Report

A thorough and engaging piece. My main point of criticism would be that the conclusion could be shortened: It’s good that you briefly restate the main points in your conclusion, but some of the conclusion is a bit too long a repetitive. Other than that, see the detailed points below.

l.52: unwanted insert

But yes, this long section could benefit from subsections.

l.57: rulerss >> rulers

l.68: It was not until… when >> It was not until… that

l.72: was the main ritual though >> was the main ritual, though

l.75: associating the heaven (tian天) with cosmos, and the earth (di地) with geography.

>> This could be somewhat expanded on, to make clearer what this shift in interpretation entails.

l.88: on >> onwards

l.91: throughout to >> delete ‘to’

l.94: Here you unexpectedly refer to ‘five rituals’. It would be good to provide some context: what were these five rituals for, etc.?

ll.94-129: It may not be immediately clear to all your readers whether second grade and medium sacrifice are the same or different? You immediately dive into a very detailed discussion of the sacrifice to the yue-zhen-hai-du, but perhaps the above point needs to be clarified first, so that readers can better understand the ritual progression.

l.141: Chen陳while >> Chen 陳, while

l.150: Here, could you specify what this difficulty was?

l.175: ranked the 47th whereas >> ranked 47th, whereas

l.176: ranked the 18th >> ranked 18th

l.209: in par with >> on a par with

l.209: Here you use the term ‘annual grand ritual’, which risks being confused with the ‘grand sacrifice’ mentioned above (l.98), which is reserved for heaven and earth. Perhaps another term could be used to avoid potential confusion.

l.226: in both political-cultural >> in both a political-cultural

l.230: than those in the >> than during the

l.236: In other word, >> In other words,

l.238: from Minister >> from the Minister

l.244: prayer-board was forbidden to be used >> the use of prayer-boards

ll.246-247: (788) when the old ritual was resumed to use prayer-boards offered by the imperial rulers in person

>> (788), when the old ritual was resumed, that the use of prayer-boards offered by the imperial rulers in person was re-introduced

l.257: filthy >> ritually polluting

l.259: at a certain direction >> in a certain direction

l.264: stair-steps >> steps

l.264: Subsequently temples >> Subsequently, temples

l.269: wooden vessel >> plural!

l.283: identified >> marked out

l.287: bowling >> bowing

l.297: The correlation of the fours with the fives is not obvious and may require some explanation on your part.

ll.305-309: Here you make a very interesting distinction between cosmographical and geographical conceptions of the four seas that changed over time. But in the whole of the preceding discussion, you did not make this highly important point. I think it would be good to clearly spell out this point above, when you discuss the evolution of the conception of the four seas over time. This way, when you mention it here, readers would be mentally prepared for this important point.

l.311: embody to >> delete ‘to’

l.328: was >> were

l.333: commissioner >> plural

l.341: see his footprints >> follow his footsteps

l.357: to the destination Guangzhou >> to his destination, Guangzhou

l.370: entitled all >> bestowed titles on all

l.378: four seas gods >> four sea gods

l.381: that was >> who was

l.382: perhaps here you could provide some of the references to the different scholarly opinions.

l.440: Unwanted insert.

l.453: new normal >> new norm

l.456: natural disasters and cultural reasons >> these two do not easily fit together without further explanation on your part.

l.461: As just studied >> As mentioned above

l.465: foreign treasure >> plural! (large number of)

l.469: on summer solstice >> on the summer solstice

l.478: who was so much into immortality >> who was so concerned with immortality

l.496: realized >> answered

l.509: Here two references in brackets follow each other. The presentation needs to be emended.

l.514: for good health >> with good health

l.516: to be designated >> not sure why this is added here, since all of them are designated.

l.523: rulings >> rules

l.524: successors. yue-zhen-hai-du >> capital after full stop

l.531: they may not survive >> they risked their lives

l.544: was held as grand as it should be >> was held in as grand a style as it should be

l.546: was echoing >> echoed

l.566: Is the South Sea God an ‘it’ or a ‘he’?

l.574: in the ancient time >> in ancient times

l.578: to be a Buddhist >> to Buddhism

ll.575-578: This statement seems slightly judgemental and polemical in the context of a Religious Studies article. As a scholarly principle, it is certainly right that you problematize the appropriation of the legend by the Buddhists. But it is equally important that you refrain from passing judgement on such a phenomenon, which after all is common among inter-religious exchanges.

l.592: robe >> robe appeared

l.593: he was declined >> his request was declined

l.610: legend above >> above legend

l.616: he had better accepted >> it was in his interest to accept

ll.619-621: it shows that the South Sea God worshiped by the state took advantage of Buddhism to change his image of an evil spirit because the legend helped the deity to get rid of his “hot-tempered” character.

>> it shows that the South Sea God worshiped by the state was able to benefit from Buddhism, in having his “hot-tempered” reputation transformed by means of this new legend.

l.621: After converted to a Buddhist, the deity was >> After converting to Buddhism, the deity became

l.624: in the legitimate way >> in a legitimate way

l.626: on par >> on a par

l.652: moderation >> do you mean modification?

l.637: the title here is much too long and fails to capture the gist of what follows

Here’s my suggestion: Systematization of ritual sacrifice to the yue-zhen-hai-du

l.664: distinct…on purpose >> purposefully distinct (remove on purpose at the end)

l.686: culture, and thoughts to study the ritual system >> and culture (the remaining part of the sentence makes little sense)

l.688: true to >> true of

l.691: throughout dynasties >> throughout the various dynasties

l.693: by different emperors >> by the different emperors

l.695: resulted a lot >> resulted in a lot

l.695: relics well preserved >> relics that are well preserved

Conclusion

I study

I aim

>> I have studied

>> My aim has been

l.705: was with >> was integrated with

l.710: in the ‘orthodox’ sense >> not clear what you mean by this; best to delete.

l.737: Different from >> Unlike

ll.738-739: because of conferring…to show >> in order to confer…and show

l.741: was sent >> who was sent

l.742: as I find evidence in evidence in >> as evidenced in

ll.772-773: Nevertheless, Buddhism can not go beyond the state to exercise the religious right.

>> This statement had best be removed, as it adds nothing to the discussion and unnecessarily opens a new topic.

Throughout: Sometimes you give references: Author, Year, Page; at other times Author Year, Page >> the comma needs to be standardize either way.

present day >> present-day

Often, Chinese characters are inserted without any spacing. This is typographically inelegant.  

Reviewer 2 Report

See attached. 

Reviewer 3 Report

This paper is a good historical study about the sacrificial ritual to the South Sea God in Tang China. Nonetheless, I’d like to mention a few things the author may want to consider while revising the paper.

  • “Introduction” is just too simple to explain the significance of this study. Can we expect to find in the paper things other than adding our knowledge about the sacrificial rituals? It seems to me that this paper is poorly contextualized in the introductory paragraphs.
  • More relevant studies, especially those on the sacrificial ritual in general, should be cited, and more primary sources associated with the Nanhai temple should be directly quoted.
  • The structure of the paper is clearly out of balance: Part 2 is simply too long when compared with other parts.
  • The author argues on pp. 8-9 that Zhang Jiuzhang was the imperial commissioner. Is there any possibility to move to the footnote the three points used to support the argument so as to avoid interruption? Another option is to make the argument shorter.
  • The “conclusions” could be rewritten in a cleaner and more concise way.
  • For 藍田玉山 on p. 7, “Mount Yu in Lantian county” is probably a better translation.
  • In 緬然萬裏路 on p.7, 裏 should be 里.
